

# Harmine stimulates proliferation of human neural progenitors

Vanja Dakic[1,2], Renata de Moraes Maciel[1], Hannah Drummond[1,2], Juliana M. Nascimento[1,3], Pablo Trindade[1] and Stevens K. Rehen[1,2]

[1] IDOR, D'Or Institute for Research and Education, Rio de Janeiro, RJ, Brazil
[2] Institute of Biomedical Sciences, Federal University of Rio de Janeiro, Rio de Janeiro, RJ, Brazil
[3] Department of Biochemistry and Tissue Biology/Institute of Biology, State University of Campinas (UNICAMP), Campinas, SP, Brazil

## ABSTRACT

Harmine is the $\beta$-carboline alkaloid with the highest concentration in the psychotropic plant decoction Ayahuasca. In rodents, classical antidepressants reverse the symptoms of depression by stimulating neuronal proliferation. It has been shown that Ayahuasca presents antidepressant effects in patients with depressive disorder. In the present study, we investigated the effects of harmine in cell cultures containing human neural progenitor cells (hNPCs, 97% nestin-positive) derived from pluripotent stem cells. After 4 days of treatment, the pool of proliferating hNPCs increased by 71.5%. Harmine has been reported as a potent inhibitor of the dual specificity tyrosine-phosphorylation-regulated kinase (DYRK1A), which regulates cell proliferation and brain development. We tested the effect of analogs of harmine, an inhibitor of DYRK1A (INDY), and an irreversible selective inhibitor of monoamine oxidase (MAO) but not DYRK1A (pargyline). INDY but not pargyline induced proliferation of hNPCs similarly to harmine, suggesting that inhibition of DYRK1A is a possible mechanism to explain harmine effects upon the proliferation of hNPCs. Our findings show that harmine enhances proliferation of hNPCs and suggest that inhibition of DYRK1A may explain its effects upon proliferation *in vitro* and antidepressant effects *in vivo*.

## INTRODUCTION

Throughout life, specific regions in the human adult brain continuously generate neural cells from a pool of neural progenitor cells (hNPCs). Many physiological and pathological events are able to control neurogenesis by modulating proliferation, differentiation, maturation and integration of newborn neurons into the existing circuitry (*Zhao, Deng & Gage, 2008*). This balance can be disrupted by chronic stress (*Egeland, Zunszain & Pariante, 2015*) depression (*Mahar et al., 2014*), aging (*DeCarolis et al., 2015*), and neurodegenerative diseases (*Winner & Winkler, 2015*).

Classical antidepressants can reverse or block stress-induced hippocampal atrophy in rodents, mostly by stimulating neuronal proliferation (*Malberg et al., 2000*). Fluoxetine, one of the most used selective serotonin reuptake inhibitors, induces proliferation of rat hypothalamic (*Chen et al., 2007*; *Sachs & Caron, 2015*; *Sousa-Ferreira et al., 2014*) and

Corresponding author
Stevens K. Rehen, srehen@lance-ufrj.org

hippocampal neural progenitors *in vitro* and *in vivo* (*Chen et al., 2007*; *Sachs & Caron, 2015*). Unfortunately, treatment with classic antidepressants leads to full remission in only 50% of patients (*Nestler et al., 2002*), causes side effects and the time required for achieving therapeutic response is usually measured in weeks. Thus, the demand for novel psychopharmacological agents able to revert depression remains significant.

Beta-carbolines, a large group of indole alkaloids are widely distributed in plants. Two members of this group, harmine and harmaline, have been found in human plasma after ingestion of Ayahuasca (*Callaway et al., 1996*), a psychotropic beverage traditionally used in the Amazonian region of South America as part of local religious ceremonies (*Labate & Feeney, 2012*).

Evaluation of the effects of a single dose of Ayahuasca in six volunteers with a current depressive episode suggested that this plant decoction has fast-acting anxiolytic and antidepressant effects (*Osorio Fde et al., 2015*). Moreover, in rodents, the use of harmine leads to the reduction of symptoms associated with depression (*Farzin & Mansouri, 2006*) and re-establishment of normal levels of hippocampal brain-derived neurotrophic factor (BDNF) (*Fortunato et al., 2009*).

Apart of these initial studies, there are no data available regarding the neurogenic effects of harmine in humans. Here we examine the effects of harmine on the proliferation of human neural progenitor cells derived from pluripotent stem cells. We show that harmine increased the pool of neural progenitor cells and that inhibition of DYRK1A is the possible mechanism involved in those proliferative effects.

# MATERIAL AND METHODS

## Chemicals

Harmine (286044), INDY (SML1011), and pargyline hydrochloride (P8013) were purchased from Sigma-Aldrich and diluted in DMSO. Subsequent dilutions were made in aqueous solution. Click-it EdU kit and BOBO$^{TM}$-3 were purchased from Thermo Fisher Scientific. All controls received an amount of vehicle equivalent to drug treatment conditions and no significant difference was observed between controls with (DMSO) or without vehicle.

## Human pluripotent stem cells

Human embryonic stem cells (*Fraga et al., 2011*) were cultured under feeder-free culture conditions on Matrigel (BD Biosciences) coated dishes (Corning) in Essential 8$^{TM}$ Medium (Thermo Fisher Scientific). Passaging was performed enzymatically using Accutase (Millipore) by splitting colonies in clumps every 4–5 days and re-plating on Matrigel-coated dishes, having their medium changed every day. All cells were maintained at 37 °C in humidified air with 5% $CO_2$.

## Human neural progenitor cells

To induce embryonic stem cells to direct neural differentiation, we performed an adaptation of *Baharvand et al. (2007)* protocol (*Paulsen et al., 2012*). Briefly, 70% confluent BR1 culture was differentiated to the neural lineage in defined adherent culture by retinoic acid and basic fibroblast growth factor (bFGF) within 18 days of culture. On the 18th day, neural

tube-like structures were collected and replated on dishes coated with 10 µg/mL of Poly-L-ornithine and 2.5 µg/mL of laminin (Thermo Fisher Scientific). The population of hNPCs that migrated from neural tube-like structures was tested for expression of neuronal markers and expanded. Expansion was done in N2B27 medium supplemented with 25 ng/mL bFGF and 20 ng/mL EGF (Thermo Fisher Scientific). N2B27 medium consisted of DMEM/F-12 supplemented with 1X N2, 1X B-27, 1% penicillin/streptomycin (Thermo Fisher Scientific). Cells were incubated at 37 °C and 5% $CO_2$. Medium was replaced every other day. The hNPCs were expanded for no more than five passages.

## High content screening

Cell proliferation, cell death and DNA damage experiments were performed in a High Content Screening (HCS) format. The hNPCs (1,500 cells/per well) were plated on a multiwell 384 µClear plate (Greiner Bio-One) coated with 100 µg/mL Poly-L-ornithine and 10 µg/mL laminin (Thermo Fisher Scientific). After 24 h, cells were treated for 4 days in quintuplicate (five wells per condition) with harmine, INDY and pargyline in N2B27 medium supplemented with bFGF and EGF. On day 4 cells were labelled with 10 µM EdU for 2 h (cell proliferation) or BOBO$^{TM}$-3 (cell death) for 30 min prior to fixation or image acquisition, respectively.

## High content analysis

All images were acquired on Operetta high-content imaging system (Perkin Elmer). For proliferation, incorporated EdU was detected with Alexa Fluor 488 using Click-iT EdU kit following manufacturer's instruction. Immunocytochemistry for Ki-67 was performed after EdU AF488 labelling. Total number of cells was calculated by nuclei stained with DAPI. S phase was determined by percentage of total cells labelled with EdU. Whereas dividing cells in all phases of cell cycle, exempting G0, were measured by Ki-67 positive cells as percentage of total cells. Images were acquired with a 10× objective with high numerical aperture (NA).

For cell death analysis, cells were labelled with a fluorophore dye cocktail, containing the cell-permeant nuclear dye Hoechst and the cell-impermeant nuclear dye BOBO$^{TM}$-3 in fresh N2B27 medium for 30 min at 37 °C and 5% $CO_2$. After incubation, the dye cocktail was replaced for new medium and live cell imaging was performed using temperature and $CO_2$ control option (TCO) of Operetta, set to 37 °C and 5% $CO_2$ at 10× magnification. For DNA damage analysis, immunocytochemistry was performed on fixed cells after 4 days of treatment using H2AX antibody, and images were acquired at 10× magnification. All quantification analysis were normalised to the number of cells in the well segmented by nucleus dyes. $H_2O_2$ was used as positive control for both cell death and DNA damage.

All analyses sequences were designated by combining segmentation steps with morphological and fluorescence based object characterizations using the image analysis software Harmony 3.5.1 (Perkin Elmer).

## Immunocytochemistry

hNPCs were fixed in formaldehyde 3.7% for 15 min at RT and permeabilized in 0.2% Triton X-100 for 15 min Primary antibodies were incubated overnight in 2% BSA at 4 °C, following 40 min of 2% BSA blockage. After washing with PBS, secondary antibodies were incubated

for 1 h at RT in the dark. Cells were washed three times with PBS and nuclei were stained with DAPI. Coverslips were mounted on slides using Aqua-Poly Mount (Polysciences) whereas cells on the 384 well plates were covered with glycerol and sealed with AlumaSeal CS (Excel Scientific) for image acquisition in confocal microscopy (Leica) and Operetta (Perkin Elmer), respectively. Primary antibodies used: mouse anti-MAP2 (Sigma-Aldrich), mouse anti-Ki-67 (BD Biosciences), rabbit anti-PAX6 (Santa Cruz Biotechnology), rabbit anti-GFAP (Dako), rabbit anti-$\gamma$-H2AX (Cell Signaling Technology), rabbit anti-FOXG1 (Abcam), rabbit anti-DYRK1A (Sigma-Aldrich), rabbit anti-SOX2 (Millipore), mouse anti-$\beta$-tubulin III (Millipore), mouse anti-nestin (Millipore), rabbit anti-TBR2 (Millipore). Secondary antibodies used: goat anti-mouse Alexa Fluor 594 and goat anti-rabbit Alexa Fluor 488 (Thermo Fisher Scientific). Data are expressed as relative protein expression in comparison with basal protein expression in control with vehicle (DMSO).

### Statistical analysis

All data are expressed as mean $\pm$ sem. Results were accepted as statistically significant at $p < 0.05$, as determined using one-way ANOVA with Tukey's multiple comparison test. A minimum of 10,000 hNPCs was counted per condition/per experiment.

## RESULTS

We generated hNPCs from human embryonic stem cells using a protocol that recapitulates the early steps of neural development (*Baharvand et al., 2007*). Characterization of specific cell markers was done on day 0 (Fig. S1). More than 90% of the hNPCs express markers of neural progenitors, such as SOX2 (sex determining region Y-box 2), nestin (intermediate filament protein; neuroectodermal stem cell marker), PAX6 (paired box 6), FOXG1 (forkhead box G1; transcriptional repressor important for development of the brain and telencephalon), TBR2 (transcription factor Eomes; key regulator of neurogenesis in the subventricular zone) (Figs. 1A, 1B, 1D, 1E and 1G). Additionally, differentiated cells expressing $\beta$-Tubulin III (a class III member of the $\beta$ tubulin protein family primarily expressed in neurons), GFAP (glial fibrillary acidic protein, marker of astrocytes), or MAP2 (marker of neuronal soma and dendrites) are also observed (Figs. 1B, 1C and 1G). Ninety percent of cells express the dual specificity tyrosine-phosphorylation-regulated kinase 1A (DYRK1A) (Figs. 1F and 1G).

### Harmine increases proliferation of human neural progenitors: a possible role of DYRK1A

Pharmacokinetic studies have shown a high variability in the bioavailability of ayahuasca alkaloids in the human plasma: 0.5–5 ng/mL (*Yritia et al., 2002*) and 36.4–222.3 ng/mL (*Callaway et al., 1996*). On the other hand, *in vitro* studies have been using a wide spectrum of harmine concentrations, ranging from 333 nM to 22.5 $\mu$M (*Gockler et al., 2009*; *Hammerle et al., 2011*; *Martinez de Lagran et al., 2012*; *Mazur-Kolecka et al., 2012*; *Wang et al., 2015*). In order to examine its effects upon proliferation, we incubated human neural cells with harmine ranging from 0.1 to 22.5 $\mu$M for 96 h (Figs. S1 and S2). The 7.5 $\mu$M concentration of harmine was the most effective on proliferation of hNPCs (Fig. S2),

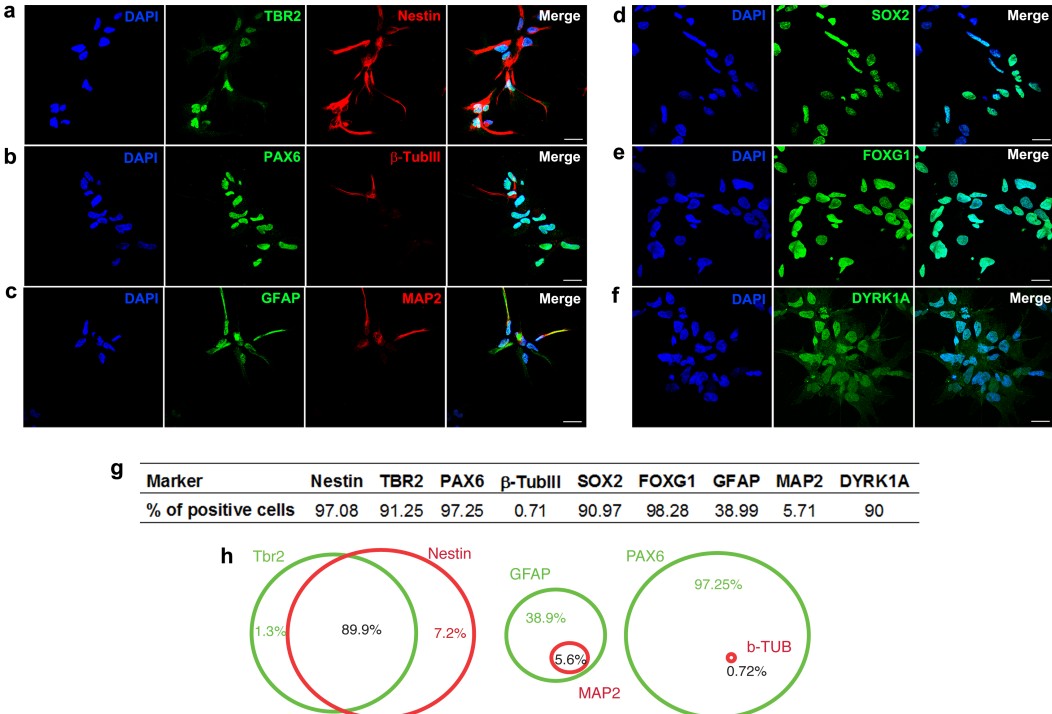

| Marker | Nestin | TBR2 | PAX6 | β-TubIII | SOX2 | FOXG1 | GFAP | MAP2 | DYRK1A |
|---|---|---|---|---|---|---|---|---|---|
| % of positive cells | 97.08 | 91.25 | 97.25 | 0.71 | 90.97 | 98.28 | 38.99 | 5.71 | 90 |

**Figure 1 Characterization of neural progenitor cells derived from human embryonic stem cells.** Representative images showing hNPCs stained for (A) TBR2 and Nestin, (B) PAX6 and β-Tubulin III, (C) GFAP and MAP2, (D) SOX2, (E) FOXG1, (F) DYRK1A. (G) Quantification of cell markers. (H) Venn diagram showing percentage of double stained cells. A minimum of 10,000 hNPCs was counted per marker. Scale bar: 25 μm.

increasing the pool of proliferating cells by 71.5% (Figs. 2A and 2B). No evidence of cell death or DNA damage in response to harmine as measured by BOBO-3 (Figs. 3A and 3B) H2AX labelling (Figs. 3C and 3D) was observed.

Treatment with harmine increased by 64.4% the specific pool of neural progenitors, which actively participates in adult neurogenesis (Nestin and GFAP labelled, Figs. 2C and 2D) (Fukuda et al., 2003). The number of cells positive for SOX2, MAP2 and FOXG1 was not altered (Fig. S3).

Harmine has been shown to be an inhibitor of DYRK1A (Becker & Sippl, 2011; Gockler et al., 2009), other DYRK family members (Gockler et al., 2009) and monoamine oxidase (MAO) (Santillo et al., 2014). To verify whether these two targets are involved in the harmine-dependent increase in neural proliferation, we examined the effect of two functional analogs of harmine in human neural cells: INDY (15 μM), an inhibitor of DYRK1A; and pargyline (10 μM), an irreversible selective inhibitor of MAO but not DYRK1A.

Harmine and INDY increased proliferation of hNPCs in comparison with control, analysed by two different methods, while pargyline did not induce the same effect. Concomitant inhibition of both DYRK1A and MAOs produced similar effect as harmine or INDY alone (Fig. 4).
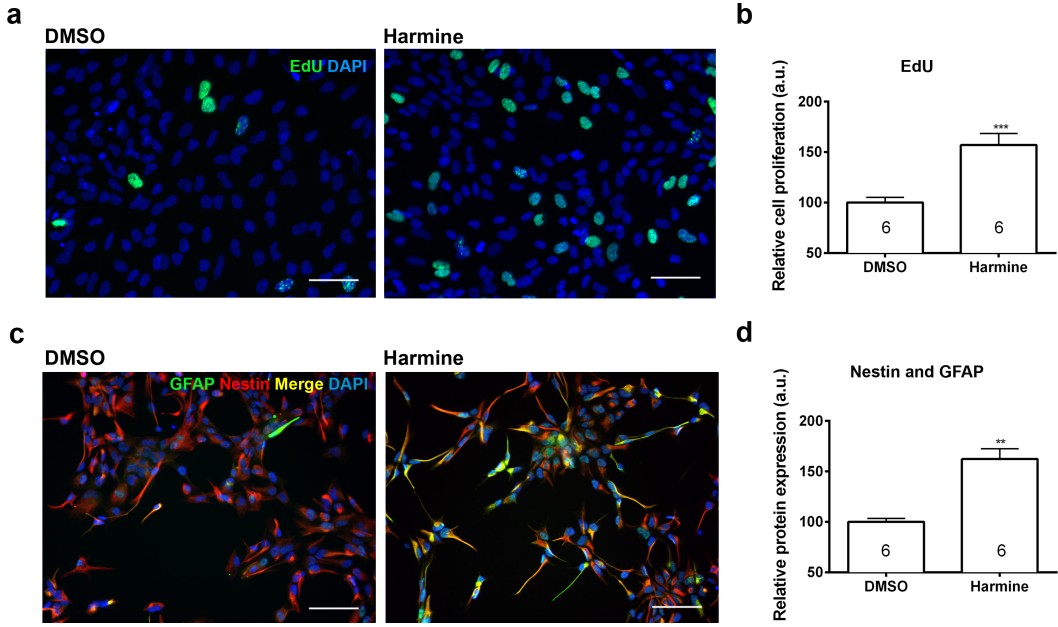

**Figure 2** **Quantification of the proliferation and differentiation of hNPCs after treatment with harmine.** (A) Representative images of EdU staining. (B) Cell proliferation (S phase) relative to DMSO control, measured by EdU incorporation. (C) Representative images of GFAP and Nestin staining. (D) Expression of Nestin and GFAP proteins relative to DMSO control. A minimum of 10,000 hNPCs was counted per condition/per experiment. Data were analysed by one-way ANOVA with Tukey's multiple comparison test, **$p < 0.001$, *** $< 0.0001$. Values represent mean ± sem. The number inside the bar represents the number of experiments in each group. Scale bar: 100 μm.

## DISCUSSION

The use of *in vitro* models can potentially clarify mechanisms related to proliferation and differentiation of newborn neurons (*Cai & Grabel, 2007*), which happens massively during embryogenesis but also later in specific brain regions. Numerous studies have shown a close relationship between disturbed adult neurogenesis and depression, highlighting the need for more efficacious and faster-acting treatments. Thus, the aim of this study was to investigate the effects of harmine, a compound with potential antidepressant properties, on human neural progenitor cells derived from pluripotent stem cells. Here we show that harmine increases proliferation of hNPCs, having inhibition of DYRK1 as the possible mechanism.

We describe that, in cell cultures containing more than 90% of hNPCs, treatment with harmine increased proliferation without DNA damage or cell death. These results are consistent with findings published by *Hammerle et al. (2011)*, in which exposure of chick embryos to harmine resulted in a strong increase in BrdU incorporation and number of mitotic cells in the spinal cord.

Interestingly, harmine treatment also increased the number of early progenitor cells, expressing both GFAP and Nestin. At first, this result suggests that harmine drives the differentiation of hNPCs into radial glial cells, which are the major source of neuronal and
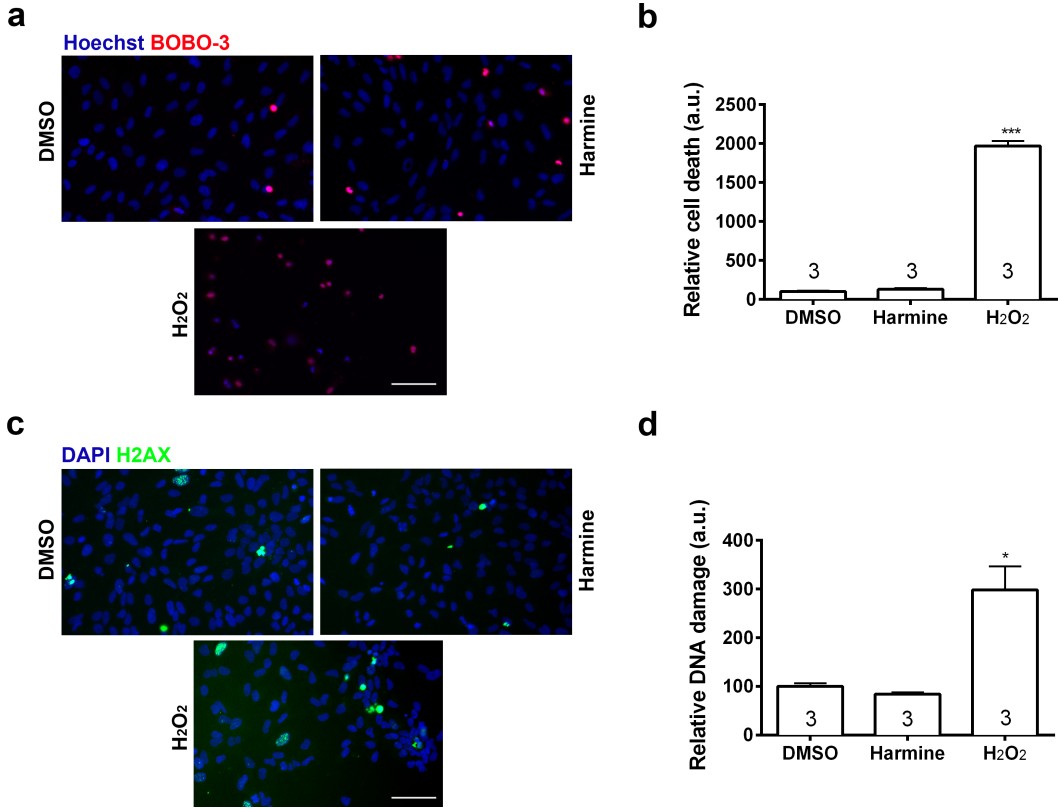

**Figure 3 Percentage of cell death and DNA damage in neural cells treated with harmine.** (A) Representative immunostaining images of BOBO-3 (red) positive cells. Nuclei are stained with Hoechst (blue). (B) Cell death relative to DMSO control, measured as the percentage of cells stained with BOBO-3. (C) Representative images of cells positive for H2AX (green). Nuclei are stained with DAPI (blue). (D) Quantification of H2AX positive cells relative to DMSO control, as percentage of cells stained with H2AX. $H_2O_2$ was used as positive control for both cell death and DNA damage. A minimum of 10,000 hNPCs was counted per condition/per experiment. Data were analysed by one-way ANOVA with Tukey's multiple comparison test, $^*p < 0.05$, $^{***} < 0.0001$. Values represent mean ± sem. The number inside the bar represents the number of experiments in each group. Scale bar: 100 μm.

glial progenitors in the developing brain (*Gotz & Barde, 2005*). Similarly, in the adult hippocampus of rodents, these neural precursors are responsible for late neurogenesis and gliogenesis (*Zhao, Deng & Gage, 2008*). In this context, levels of GFAP tend to increase in cells that shifted to a glial fate. On the other hand, despite being GFAP+ cells, at this developmental stage, these progenitors can also shift to a phenotype PSA-NCAM+, which gives rise to neurons (*Fukuda et al., 2003*). In the light of this evidence, we suggest that harmine potentiate proliferation of radial glia-like cells (GFAP+/Nestin+) derived from hNPCs, which are capable to generate both neurons and astrocytes. Further experiments should be done in order to confirm the terminal fate of harmine-treated hNPCs.

Harmine has been described as an inhibitor of MAO and DYRK1A (*Gockler et al., 2009*; *Santillo et al., 2014*). MAO inhibition increases serotonergic neurotransmission in the adult brain (*Finberg, 2014*), which is a key component of the classical antidepressant action. The main outcome of treatment with MAO inhibitors is upregulation of cell proliferation

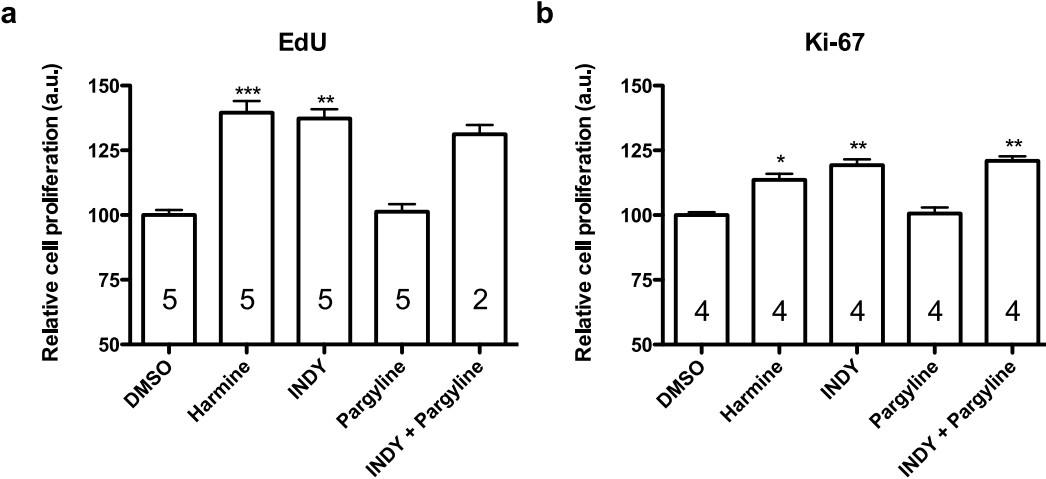

**Figure 4** **Quantification of EdU and Ki-67 labelling in human neural progenitor cells in response to harmine and its functional analogs.** (A) Cell proliferation relative to DMSO control, measured by EdU incorporation. (B) Cell proliferation relative to DMSO control, measured by Ki-67. Values represent mean ± sem. A minimum of 10,000 hNPCs was counted per condition/per experiment. Data were analysed by one-way ANOVA with Tukey's multiple comparison test, $*p < 0.05$, $**p < 0.001$, $*** < 0.0001$. Values represent mean ± sem. The number inside the bar represents the number of experiments in each group.

and neurogenesis in the hippocampus (*Manev et al., 2001*). Furthermore, it has been reported that mice lacking MAO presented reduced proliferation of neural progenitor cells (*Cheng et al., 2010*). In order to verify if MAO inhibition plays a role in harmine-induced proliferation we used the specific MAO inhibitor, pargyline. Pargyline did not alter the levels of harmine-induced proliferation. While these results indicate that MAO is not involved in the proliferative effects of harmine, we cannot rule out that the lack of effect of pargyline is due to the absence of MAO in neural progenitor cells In fact, MAO-dependent changes in NPCs proliferation were described late in embryonic development, mediated by serotonin (*Cheng et al., 2010*).

On the other hand, treatment with INDY, an inhibitor of DYRK1, induced proliferation similarly to that observed with harmine. These results are consistent with the findings of Wang and collaborators, where harmine-dependent DYRK1A inhibition increases the proliferation of pancreatic beta cells (*Zhou et al., 2015*). Since both harmine and INDY have multiple targets on DYRK family, including regulators of cell cycle (*Becker & Sippl, 2011*), we should consider other potential players of INDY- and harmine-mediated increase in proliferation.

INDY can also inhibit DYRK1A closely related kinases, DYRK1B and DYRK2. In cancer studies DYRK1B and DYRK2 have been described as modulators of proliferation (*Adayev, Wegiel & Hwang, 2011*; *Zhou et al., 2015*). In fact, harmine also inhibits DYRK1B and DYRK2, but the efficiency of this inhibition is, respectively, 5- and 50-fold lower in comparison to DYRK1A (*Adayev, Wegiel & Hwang, 2011*; *Bain et al., 2007*; *Gockler et al., 2009*). Thus, DYRK1A emerges as the major candidate for mediating the increase in proliferation seen in this study. Also, it was shown that DYRK1A directly phosphorylates p53 and leads to the induction of p53 target genes, attenuating proliferation of rat and

human neural progenitor cells (*Park et al., 2010*). Further studies are needed to reveal other aspects of human cell proliferation stimulated by harmine.

It is worth to mention that DYRK1A can also modulate migration (*Pons-Espinal, Martinez De Lagran & Dierssen, 2013*) and differentiation (*Kurabayashi & Sanada, 2013*; *Yabut, Domogauer & D'Arcangelo, 2010*) of neural progenitors in rodents. Neurogenesis contemplates the interplay of proliferation, migration and differentiation. While our data show that harmine stimulates proliferation of hNPCs, further studies upon migration and differentiation may clarify whether neurogenesis (or gliogenesis) are indeed increased by harmine in humans.

Taken together our results suggest that harmine exert proliferative effects in human neural progenitors, particularly in radial glia-like cells (GFAP+/Nestin+), by inhibiting DYRK1A These findings shed light on the possible mechanisms behind the antidepressant effects of Ayahuasca described in patients.

### Abbreviations

| | |
|---|---|
| **hNPCs** | Human neural progenitors cells |
| **DYRK** | Dual-specificity tyrosine phosphorylation-regulated kinase |
| **MAO** | Monoamine oxidase |
| **GFAP** | Glial fibrillary acidic protein |
| **MAP2** | Microtubule-associated protein 2 |

## ACKNOWLEDGEMENTS

This work is part of the PhD thesis of VD. We thank Ismael Gomes, Marcelo Costa and Igor Lima da Silva for technical assistance and Dr. Mauro de Freitas Rebelo for assistance with statistical analysis.

### Funding

This study was funded by the following Brazilian funding agencies: National Council for Scientific and Technological Development (CNPq), Foundation for Research Support in the State of Rio de Janeiro (FAPERJ), Coordenação de Aperfeiçoamento de Pessoal de Nível Superior (CAPES), Funding Authority for Studies and Projects (FINEP), Brazilian Development Bank (BNDES) and São Paulo Research Foundation (grant 2014/21035-0). The funders had no role in study design, data collection and analysis, decision to publish, or preparation of the manuscript.

### Grant Disclosures

The following grant information was disclosed by the authors:
National Council for Scientific and Technological Development (CNPq).
Foundation for Research Support in the State of Rio de Janeiro (FAPERJ).
Coordenação de Aperfeiçoamento de Pessoal de Nível Superior (CAPES).

Funding Authority for Studies and Projects (FINEP).
Brazilian Development Bank (BNDES).
São Paulo Research Foundation: 2014/21035-0.

## Competing Interests

The authors declare there are no competing interests.

## Author Contributions

- Vanja Dakic conceived and designed the experiments, performed the experiments, analyzed the data, wrote the paper, prepared figures and/or tables, reviewed drafts of the paper.
- Renata de Moraes Maciel conceived and designed the experiments, performed the experiments, analyzed the data, prepared figures and/or tables, reviewed drafts of the paper.
- Hannah Drummond performed the experiments, reviewed drafts of the paper.
- Juliana M. Nascimento prepared figures and/or tables, reviewed drafts of the paper.
- Pablo Trindade wrote the paper, reviewed drafts of the paper.
- Stevens K. Rehen conceived and designed the experiments, contributed reagents/materials/analysis tools, wrote the paper, reviewed drafts of the paper.

## Data Availability

The raw data has been supplied as Data S1.

## Supplemental Information

Supplemental information for this article can be found online at http://dx.doi.org/10.7717/peerj.2727#supplemental-information.

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
