# Peer review of "Harmine stimulates proliferation of human neural progenitors"

_PeerJ, doi:10.7717/peerj.2727_

## Round 0.1 · original submission · Minor Revisions

Dear Dr Rehen,

Having considered the reviewers comments I would like you and co-authors to please address the minor comments raised by the reviewers before a final decision is made on the acceptance of your manuscript.

In particular please address Referee 1's second and last comments. and Referee 2's comments 9,10 and 11.

I look forward to receiving your revised manuscript. Please attach a letter detailing the changes made to the manuscript addressing each of the reviewers comments.

Vasanta Subramanian

Reviewer 1 ·

Basic reporting

The manuscript meets the basic reporting criteria established by the Journal.

Experimental design

The experimental design and methods are adequate.

Validity of the findings

In general I consider the findings valid. I only have some doubts regarding the images shown in Figure 2c. They do not visually reflect the quantitative data shown in the accompanying bar graph (Figure 2d).

Additional comments

Manuscript: PeerJ #11527
Title: "Harmine stimulates proliferation of human neural progenitors"

The present manuscript addresses the capacity of harmine, a plant alkaloid present in the Amazonian psychoactive beverage ayahuasca, to stimulate proliferation of neural progenitor cells in vitro.

This a stimulating field of research and I consider this paper timely and important. Ayahuasca is receiving increased attention from the biomedical research community due to its therapeutic potential in the treatment of depression and addiction. I recommend its publication after some minor changes.

I would like to make the following comments / recommendations:

- Line 55: the term “human models of neurogenesis” sounds odd. Maybe the authors could write instead “there are no data available regarding the neurogenic effects of harmine in humans”

- Figure 2c-2d: while the bar graph (2d) shows clear increases in Nestin and GFAP levels after harmine, the images chosen to illustrate this effect (2c) are quite ambiguous, i.e., not much difference can be inferred visually between the DMSO and harmine pictures. Maybe the authors could comment on the discrepancy or use other representative images..

- I find the last paragraph (262-266) a bit too far-fetched considering the data reported. Proliferation is a necessary but not sufficient condition to establish potential neurogenesis or gliogenesis. I suggest that they tone it down.

- Related to the comment above, a limitations paragraph should be included. It should be stated that before neuro or gliogenesis can be established, migration and differentiation experiments should also be conducted.

Reviewer 2 ·

Basic reporting

The manuscript is well written and effectively describes relevant results in a self-contained manner. The rationale for the experiment is clear, and the data presentation is appropriate, with quite informative figures and statistical analyses. Minor concerns:

1) Abstract: The B-carboline with the highest concentration? Or a B-carboline present in high concentration?

2) Abstract: The pool of proliferating hNPCs increased by 57% or by 71.5%? One of the 2 abstracts included in the documentation has a typo.

3) Line 38: hippocampal as well hypothalamic, please make explicit what is what.

4) Line 47: please cite appropriate references for the ritual use of Ayahuasca.

5) Line 65: no SIGNIFICANT difference.

6) Line 90: 1,500 cells with comma.

7) Line 120 and elsewhere: Be consistent in abbreviations (h for hour, min for minutes).

8) Line 132 and elsewhere: Be consistent in product details and abbreviations (USA, UK, display city and state for all products, or for none).

9) Line 139: How were the data distributed? Was each group sample drawn from a normally distributed population? Did all populations have a common variance? Were all samples drawn independently of each other? Within each sample, were the observations sampled randomly and independently of each other?

10) A Venn diagram would greatly improve Figure 1.

11) Line 164: How does the range of concentrations investigated compare to the levels attained in the brain after oral consumption of Ayahuasca?

12) Line 178 and elsewhere including Supplementary: Data WERE analyzed.

13) Line 223: Hammerle and other Hammerle et al, consider citing only once.

14) Line 224: Consider "exposure of chick embryos to harmine"

15) Lines 240/241: not only setotonin, add reference.

16) Line 257: the major candidate FOR mediating.

Experimental design

Excellent experimental realization and documentation, important and timely questions.

Validity of the findings

The results are quite convincing and the discussion is compelling. On lines 243/244, consider stating your hypothesis for the lack of proliferation induced by pargyline.

---

## Round 0.2 · accepted · Accept

Your revised manuscript has now been seen by both reviewers and they recommend acceptance for publication. I am now happy to accept the manuscript for publication in PeerJ

Reviewer 1 ·

Basic reporting

No comments

Experimental design

No comments

Validity of the findings

No comments

Additional comments

The authors have adequately addressed all my minor concerns. I congratulate them for this interesting paper and recommend its publication in its present form.

Reviewer 2 ·

Basic reporting

OK

Experimental design

OK

Validity of the findings

OK

Additional comments

I'm satisfied with the response of the authors and the changes made to the manuscript.